# Intersection and Considerations for Patient-Centered Care, Patient Experience, and Medication Experience in Pharmacogenomics

**DOI:** 10.3390/pharmacy11050146

**Published:** 2023-09-14

**Authors:** Logan T. Murry, Lisa A. Hillman, Josiah D. Allen, Jeffrey R. Bishop

**Affiliations:** 1College of Pharmacy, The University of Iowa, Iowa City, IA 52242, USA; 2College of Pharmacy, The University of Minnesota, Minneapolis, MN 55455, USA; hill0667@umn.edu (L.A.H.); or alle0861@umn.edu (J.D.A.); jrbishop@umn.edu (J.R.B.); 3Department of Pharmacy, St. Elizabeth Healthcare, Edgewood, KY 41017, USA; 4Medical School, The University of Minnesota, Minneapolis, MN 55455, USA

**Keywords:** patient-centered care, pharmacogenomics, patient experience, medication experience

## Abstract

As healthcare continues to embrace the concept of person- and patient-centered care, pharmacogenomics, patient experience, and medication experience will continue to play an increasingly important role in care delivery. This review highlights the intersection between these concepts and provides considerations for patient-centered medication and pharmacogenomic experiences. Elements at the patient, provider, and system level can be considered in the discussion, supporting the use of pharmacogenomics, with components of the patient and medication experience contributing to the mitigation of barriers surrounding patient use and the valuation of pharmacogenomic testing.

## 1. Introduction: Patient-Centered Care, Pharmacogenomics, and Patient Experience

Over the past several years, pharmacy practice and healthcare as a whole have emphasized the importance of a shift toward the delivery of patient and/or person-centered care (PCC) [1,2]. The concept of PCC, which is at the center of the Joint Commission of Pharmacy Practitioner’s Pharmacist’s Patient Care Process [3], “integrates the preferences, values, and beliefs of the person into the process of decision-making, producing a treatment plan that is both appropriate and meaningful for the patient while supporting the role of patients making informed and active choices, rather than remaining passive recipients of their care” [1,4]. PCC has been shown to provide several advantages to the patient and healthcare provider, including improved disease control, improved treatment adherence, reduced patient anxiety, and increased patient engagement [5,6]. In an umbrella review of articles defining and implementing PCC by Grover et al. [1], common elements of PCC were identified on the patient, provider, and system levels. Articles identified in the review by Grover et al. emphasized the importance of patient-provider relationships, communication, and patient empowerment and support [1]. 

Alongside PCC, there is a growing appreciation for multimodal approaches to personalized care and identifying optimal medication regimens based on each patient’s unique clinical presentation, environmental factors, metabolic and other physiologic processes, therapeutic drug monitoring, and, most recently, the implementation of pharmacogenomics (PGx). Existing evidence supports the clinical application of PGx for drug selection and dosing for a wide variety of medications, including but not limited to antidepressants, antiplatelets, opioid analgesics, proton pump inhibitors, and warfarin [7,8,9]. Additionally, PGx cascade testing (PhaCT—the identification of potential at-risk family members for testing) has been proposed as a tool to increase medication adherence by reducing adverse drug events [10,11]. To date, PGx services have been developed and implemented in a wide variety of clinical settings, including community pharmacies and primary care [12,13,14]. In addition to the increased emphasis on the use of PGx testing, a number of studies have focused on evaluating PGx testing services using implementation science frameworks to assess for successful delivery and sustainability. For example, a study by the IGNITE Pharmacogenetics Working Group used best-worst scaling to evaluate the constructs of an implementation science framework that were deemed most important to PGx testing implementation at 17 healthcare organizations in the USA, finding that patient needs and resources constituted the most important relative construct for PGx implementation [15,16].

The identification and evaluation of common PCC elements and the institutional importance of patient needs and resources for the successful implementation of PGx in the clinical setting suggest that considering PGx patient experience and PGx medication experience may provide a holistic view of factors to consider when developing and implementing sustainable PGx programs in healthcare and pharmacy practice. According to the Agency for Healthcare Research and Quality, patient experience “encompasses the range of interactions that patients have with the healthcare system, including their care from health plans, and from doctors, nurses, and staff in hospitals, physician practices, and other healthcare facilities” [17]. A systematic review of evidence by Doyle, Lennox, and Bell identified positive associations between patient experience and clinical safety and effectiveness [18]. The medication experience has been defined by Shoemaker and Ramalho de Oliveira as an “individual’s subjective experience of taking a medication in his daily life” [19]. The authors describe it as “a practice concept that serves to understand patients’ experiences” and “medication-taking behaviors in order to meet his or her medication-related needs.” Of note, several tools and measurements have been developed to quantitatively capture and assess patients’ experience with medications and intervene with a systematic and structured approach [20,21,22]. As a key step in maximizing patient care and medication optimization through patient-centered PGx services, understanding elements of patient experience and medication experience that may contribute to PGx intervention development and implementation is essential. Herein, we provide an overview of the intersection of and considerations for PCC, patient experience, and medication experience in PGx. 

## 2. Patient Perspectives, Needs, and Resources in PGx

Patient expectations and experiences surrounding PGx are varied and often influenced by patient- and system-specific factors. A study by Bright, Worley, and Porter (2021) identified four themes surrounding patient experience with PGx testing in the community pharmacy setting: Trust, Experience, Risk/Benefit, and Clarity [23]. In this study, patients’ views surrounding PGx were influenced by trust in the healthcare system or a specific relationship as well as personal or familial experience with PGx. In their study, Bright, Worley, and Porter described that an appropriate explanation and access to additional information when discussing PGx were important factors that positively contributed to patient views of PGx, in addition to patient understanding of how information obtained from PGx testing may provide benefits [23]. Additionally, facilitating patient understanding of risks surrounding how information obtained from PGx testing might be used positively contributed to patient views on PGx [23]. Similarly, Jarvis et al. (2022) evaluated the impact of a comprehensive medication management program that included PGx testing, focusing on elements and factors associated with the patient experience. In this study, several factors associated with the patient experience were identified as potentially beneficial to care delivery, including close interaction with pharmacists, education to help patients understand the potential impact of genetics on medication response, completing PGx testing from home, as well as program privacy, ease, and security [24]. Given these findings, it is apparent that patient-specific factors, such as preferences and expectations, play an essential role in PGx services and testing. 

Importantly, the types of healthcare interactions, education, and service processes, in addition to patient preferences for them, may vary considerably within and between patient groups when using or experiencing PGx services or testing. For example, Zhang et al. evaluating a sample of Minnesota residents, found that 84% felt comfortable getting a PGx test for clinical care. In this study, the acceptability of a PGx database was positively associated with a younger age, higher education, higher health literacy, having health insurance, and prior genetic testing experience [25]. Saulsberry et al. evaluated underrepresented patient views and perceptions of personalized medication treatment through PGx [26]. Compared to White patients in this study, Black patients were less confident about whether their providers made personalized treatment decisions and were more receptive to the idea of personal genetic information playing a greater role in their clinical care. Further, compared to White patients, Black patients reported initiating discussions surrounding the impact of personal/genetic makeup with their provider less frequently, indicating the importance of enhanced communication, tailoring communication strategies, and developing support tools to account for variations in patient and medication experience in underrepresented groups [26]. 

When considering the educational elements of the patient experience with PGx, existing studies have identified counseling and patient education needs that may be required to improve the patient experience with PGx. A study by Martin et al. explored patients’ perspectives of a pharmacist-provided pharmacogenomics service, concluding that there is a need to establish pre-genotyping expectations and individualized patient education, facilitate collaboration with patient’s providers, and sustainably update patients’ PGx information over time [27]. Similarly, a scoping review by Allen, Pittenger, and Bishop identified 5 themes and 22 subthemes that reflected knowledge gaps, misunderstandings, and patient concerns regarding PGX, which could be addressed through pre- and post-test counseling [28]. Pre- and post-test themes that were suggested to be addressed in this study included addressing reasons for testing and perceived benefits, interpreting results, addressing the psychological response, and discussing a plan for follow-up and patient concerns. A particular challenge noted by the authors is the varying levels of desired information, which are dynamic over time and within individuals, depending on contextualizing factors. As an extension of these findings, the Minnesota Assessment of Pharmacogenomic Literacy (MAPL) was subsequently developed and validated as a useful tool to quantify patient knowledge across these important domains of patient experience with PGx services and testing [29]. 

## 3. Medication Experience within PGx Patient Experience

Findings from the existing literature exploring patient perspective and experience with PGx services and testing reflect the multifaceted nature of the PGx patient experience, with solutions focusing predominately on patient education and counseling. However, patient experiences of PGx are varied and extend well beyond typically studied health-related outcomes and the interventions designed to improve them. The characteristics of patients’ experiences with medications themselves may provide additional considerations that are important to the patient when determining factors impacting the delivery of PGx in a patient-centered manner. A holistic perspective of medication use that recognizes experiences with medications in everyday life, both within and beyond the context of the healthcare encounter, can provide insights into the psychosocial benefits that PGx may provide for patients. A recent concept analysis of the medication experience by Hillman et al., which expands the usefulness of medication experience in understanding the patient perspective with medications, includes six attributes that describe the nuanced components of the patient-medication relationship: ambivalence, vulnerability, socially constructed, pragmatic, contextual and nuanced, and an active and on-going process [30]. All attributes and elements of the medication experience identified in this study are included in Table 1.

Notably, the impact that medications have on patients’ lives is experienced concretely, understood pragmatically, and elicits feelings or concerns that demonstrate the vulnerability that patients may perceive or experience when considering the use of medications. With patients’ feelings of concern and vulnerability surrounding medication use in mind, PGx has the potential to provide a tangible and concrete explanation to an otherwise ambiguous medication experience for some patients. PGx services and testing results can be used to guide therapy decisions and support recommendations from providers through improved patient engagement and shared decision-making. A case study published in Current Psychiatry showed how PGx testing was used successfully to encourage a patient who had been fixated on medication therapy for her depression to consider engaging in psychotherapy, which ended up being a major component of her clinical improvement [31]. PGx testing served as a tool facilitating a bridge between evidence-based decision-making and patient understanding to optimize medication use. This example highlights the potential utility of PGx services and testing to the medication experience as described in another study by Waldman et al., who noted that PGx had a personal utility for PGx testing, even when the information was not used to inform medication recommendations. Participants in the study by Waldman and colleagues expressed the value they placed on the information and knowledge acquisition provided by PGx [32]. This insight was further described in a study by Lemke et al., who identified that approximately 60% of participants agreed that “PGx testing was helpful to me in my healthcare decision-making at this time”; yet 87.5% were satisfied with their decision to take the PGx test [33]. From these results, 27.5% of the patients in the study identified that PGx testing had value beyond the immediate application to medication therapy selection and healthcare decision-making. These results suggest that patients value information provided by PGx services and testing that contributes to their medical evaluation and further demonstrate the psychosocial benefits (i.e., peace-of-mind, vulnerability) patients experience outside the purview of the healthcare system with regards to the value of PGx testing. 

Developing and implementing PGx services while focusing on PCC and holistic patient and medication experience factors/perspectives has the potential to improve PGx services and, with them, patient experience. While not specific to PGx, a study by Gonzalez-Bueno explored the effects of a patient-centered prescription model in patients with multimorbidity, with the intervention including four specific stages or steps: (1) Patient-centered step, (2) Diagnosis-centered step, (3) Medication-centered step, and (4) Therapeutic plan. In the patient-centered step, individual therapeutic goals were discussed, and a qualitative assessment of medication adherence was performed, identifying opportunities for additional intervention based on non-adherence determinates. Additionally, steps focused on diagnosis, medication, and therapeutic plans addressed the individuals’ condition, medication regimen, and therapeutic plan, collectively considering therapeutic goals, medication risks/benefits, and optimal therapeutic plans [34]. This approach had significant positive effects on the proportion of adherent patients, mean PDC, and reduction in the number of long-term medications and hyper-polypharmacy [34]. Taking recommendations for patient education and counseling identified in the existing PGx literature and using a structured PCC approach may help to facilitate optimal PGx patient experiences, which in turn may impact desirable therapeutic and patient-management outcomes. 

While patient education and counseling are important components of the patient and medication experience related to PGx, it is critical to consider and evaluate other contributors to the patient experience in addition to patient-level factors, specifically provider- and system-level factors. As described in the review by Jarvis et al. (2022), location, privacy, and ease were important environmental elements of the patient experience in a pharmacogenomic-enriched comprehensive medication management program [24]. Additional research using service design and structured patient experience methodologies may help to capture additional elements of PGx programs and services that contribute to the patient experience. By focusing on holistic elements contributing to the patient experience within PGx services and testing, optimal interventions may be developed and implemented, with the potential to guide therapeutic decision-making and improve health outcomes. 

Importantly, addressing PCC, patient experience, and medication experience within the context of PGx services and testing may provide benefits for clinical outcomes, most notably helping to facilitate medication adherence by addressing several elements at the core of the patient’s medication experience. A study by Haga et al. suggests that patients’ increased confidence and positive attitudes towards their medication post-PGx testing may contribute to improved medication adherence [35,36]. Several studies have evaluated the impact of PGx testing on adherence to statin therapy, finding positive associations between PGx testing and adherence [37,38]. Further, existing work has shown that the use of PGx testing results during prescribing had a positive, significant effect on patient recall of physician medication recommendations, which has the potential to contribute to a number of patient and medication experience components contributing to medication adherence, such as patient engagement [39]. Finally, an evaluation of the PGx effect on medication adherence for antidepressants is currently underway [40].

## 4. Relationship Amongst PCC, Patient Experience, Medication Experience in PGx

To fully understand how patient and medication experiences relate to PCC in PGx, it is important to consider how components of these models and ideas relate. At the patient level, the existing PCC literature emphasizes the importance of patient empowerment, promoting health literacy, and involving family members to provide support and facilitate engagement in care. These concepts overlap with factors identified within the PGx patient experience literature, which emphasize the importance of access to PGx information and individualized education, patient support, and the contributions of patient-specific factors to the PGx experience. Importantly, these individual factors and education requirements at the center of the patient PGx experience relate to and potentially influence individual confidence, attitudes, and ambivalence related to medication use and experience, which may ultimately contribute to medication adherence. 

At the provider level, characteristics such as communication skills, knowledge, empathy, and respect were facilitators of PCC. The identified studies and literature placed particular emphasis on the biopsychosocial elements of PCC; communication focused on each patient’s individuality and unique needs as well as taking a holistic approach to communicating were PCC facilitators, rather than medicine-related information gathering. In both the patient and medication experience literatures, the medication experience is described as being socially created, with emphasis on co-created experiences between patient and providers. Similarly, the PGx patient experience literature emphasizes the importance of provider knowledge and skills related to PGx and how these skills and this knowledge facilitate patient trust. 

Finally, care facilities where longitudinal relationships could be developed between patients and providers, rather than acute care settings and physical environments with privacy, were facilitators of PCC. Similarly, patients reported optimal experiences with PGx services when security, privacy, ease, and convenience were components of the PGx service. This emphasizes the importance of the environment and healthcare setting on PCC and patient experience, in addition to the relationship that patients have with their providers within these institutions. Additionally, considering how the medication experience can be standardized (by, for instance, employing the Patient Centered Prescription Model and pre- and post-counseling), integrating processes to sustainably update patients’ PGx information over time and considering the context of care delivery may facilitate positive medication experiences within PGx and facilitate PCC. The intersections between PCC, patient experience, and medication experience factors are displayed in Table 2. 

## 5. Future Directions 

Future work should continue to explore the relationship between PCC, patient experience, and medication experience. The relationship between these factors and their application within PGx-informed pharmacy and healthcare interventions have the potential to address a wide variety of individual, organizational, and intervention-specific barriers that may impede patient uptake and trust in PGx services. More specifically, future work designing and evaluating PGx services and interventions should not only consider, but incorporate, factors identified within PCC, patient experience, and medication experience. Considering these factors may optimize PGx services to meet patient needs, preferences, and expectations. Finally, PCC, patient experience, and medication experience factors should be explored not only in the general context of PGx testing, but in the context of specific conditions and treatment options. Various PCC, patient experience, and medication experience factors may vary due to conditions or medication-specific personal or societal factors and should be considered and explored in greater detail moving forward. More work is needed to understand how the relationships between the factors of PCC, patient experience, and medication experience ultimately contribute to patient care and experiences with PGx.

## 6. Conclusions

PGx is a cornerstone of precision medicine and shows a number of potential benefits for improving the treatments received by patients as well as the experiences that they have in relation to their care. There are many holistic and biopsychosocial considerations that are contributing factors to the patient and medication experience. In order to provide patient-centered PGx services, providers and organizations can consider ways to address the overall patient experience and medication experience, including how information is delivered to patients, how providers are trained to provide patient-centered PGx services, and the access to and environments where care is provided. While biological precision in medicine holds promise in terms of its ability to streamline safe and effective treatments, the implementation of PGx requires that attention be paid to the core PCC elements, in addition to intersections across PCC, patient experience, and medication experience. While additional work exploring the relationship between PGx, patient experience, and medication experience factors is necessary for patient experience and the perception of PGx, considering these factors in relation to PGx service design and assessment may allow patients to fully enjoy the value and promise that pharmacogenomics has to offer. 

## Figures and Tables

**Table 1 pharmacy-11-00146-t001:** Attributes and elements of the medication experience from work by Hillman et al. [30].

Attributes	Elements of Attributes
Ambivalence	Resistance
	Necessary Evil
	Cost and Benefit
Vulnerability	Perceived and Actual Effect of Drug on Body
	Long-Term Use
	Reliance/Dependence on Healthcare System and Providers
	Reliance/Dependence on Communication and Information
Socially Constructed	Medications as Symbols
	Norms, Perceptions, Beliefs
	Social Environment Influence
	Healthcare Context and Biomedicine
	Sense of Self
Pragmatic	Ability to Evaluate from the Patient Perspective
	Priority of Wanting to Feel Well
	Barriers to Everyday Living
	Practicalities of Medication Use
Contextual and Nuanced	Illness Experience and Health Context
	Daily Life Circumstances
	Specific Medications Personal Beliefs/Attitudes/Desire for Involvement
Active Ongoing Process	Resistance and Acceptance
	Evaluative Process
	Control and Self-Regulation
	Process That Takes Time and Has No End
	Burdensome and Requires Effort

**Table 2 pharmacy-11-00146-t002:** Intersections of PCC, patient experience, and medication experience factors in PGx.

PCC Factors Identified by Grover et al. [1]	Patient and Medication Experience Factors Identified within PGx Literature
**Patient** Patient-Tailored Education and Care Patient Engagement Patient Empowerment Patient Activation, Motivation, and Education Family/Caregiver Involvement and Support	**Patient Experience** Access to PGx Information and Individualized Education Outreach to Support Patient PGx Understanding and Expectations Patient-Specific Factors (i.e., age, past PGx experiences)
	**Medication Experience** Social Environment and Influence Confidence, Positive Attitudes, and Medication Adherence
**Provider** Social Characteristics, Confidence, Knowledge, and Skills Patient Behavior Change Techniques Holistic and Biopsychosocial Approach Shared Decision-Making Communication	**Patient Experience** Patient–Provider Relationship and Trust Appropriate PGx Explanation Understanding PGx Risks and Benefits
**System** Care Coordination Organizational Structure and Systems-Level Approach	Patient–System Relationship and Trust Security, Privacy, Convenience, and Ease of PGx Testing Programs **Medication Experience** Reliance/Dependence on Healthcare System and Providers Co-constructed Experience Informational needs dynamic over time **Patient Experience** Facilitate collaboration with patient’s providers and sustainably update patients’ PGX information over time.
	**Medication Experience** Healthcare Context and Biomedicine Reliance/Dependence on Healthcare System and Providers Patient-Centered Prescription Model-Care Delivery

## Data Availability

No new data were collected in this review.

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
