# Peer review of "Intersection and Considerations for Patient-Centered Care, Patient Experience, and Medication Experience in Pharmacogenomics"

_pharmacy, 2023, doi:10.3390/pharmacy11050146_

Round 1

Reviewer 1 Report

Dear authors,

I have read the manuscript pharmacy-2525470 thoroughly. The submitted manuscript is a review paper on pharmacogenomics. Overall, I believe the quality of the text is more than sufficient for the paper to be published in "Pharmacy", and may be of reasonably high interest for the readers. I have to say that the level of English used throughout the manuscript is superb and enables the reader to follow the story without additional problem caused by so-often poor presentation.

Abstract gives all the necessary information about the contents of the paper, keywords are appropriately chosen and all the necessary literature is listed as required. There is no excessive self-citation in literature, which is suitably chosen.

Introduction gives all the necessary information for the readers. It may be a bit longer and too detailed, but I do not insist on its shortening.

Materials and methods section gives sufficiently described and are well supported. The methodology of the research is sound. Results are clearly presented and supported with large number of Tables. The same is true with Discussion. All parts are written in such a detail, that I have no further suggestions.

I suggest the manuscript is accepted for publishing in its current state.

Best regards,

Author Response

Please see attahced document.

Reviewer 2 Report

The authors present a very interesting manuscript covering aspects of the past evolution of the idea of pharmacotherapy. The review is not a place for discussion, the manuscript is very interesting and interesting also for practitioners of many specialties.
I believe that the authors have omitted a very important issue in their considerations. I mean personalization of therapy. In today's greedy, there are few clinical departments in which doctors operate, but practitioners modifying therapy based on TDM (in our opinion, it must be a practitioner, not just an interpreter of results, in addition, he must examine and assess the patient himself - only then it makes sense ). Anyway, there are already a number of summaries on this, eg 20 years observational study - digoxin, carbamazepin. Evaluations are conducted on patients with potential interactions and high risk (e.g. DOAC and nintedanib), interactions with food (e.g. DOAC and food, everything is known about VKA...), or not assessed in clinical trials, e.g. extremely high or low body weight and DOAC treatment, moreover, for DOAC, 8-year therapy evaluations are already documented. The actions taken (off-label) are often aimed at offering the patient not only an optimal, but also a more tailored treatment regimen. This, of course, affects the quality of the proceedings, but it takes people and time (and good intentions). It seems that a short reference to personalized therapy/TDM in your work makes a lot of sense because it is an intermediate stage and functions in real, everyday medicine. I believe that the initial part of the work should be modified accordingly, as well as mention in further parts and refer to it in the conclusion. I would also suggest that the conclusions should be more practical, not a summary but, above all, a road sign on the map of pharmacotherapy.

Reviewer 3 Report

Decades of research on the human genome have provided us with knowledge that can be used to increase the effectiveness and safety of treatment. Genetic information is even essential for the use of molecularly targeted drugs, which will only be effective in the presence of specific allelic variants. In turn, in the case of diseases such as depression or schizophrenia, with a significant risk of treatment failure, pharmacogenetics and pharmacogenomics provide valuable tools that enable faster selection of effective drugs. It is also worth emphasizing that the result of the genetic test does not become outdated, which means that such a pharmacogenetic test is enough to be performed once in a patient's lifetime. In this way, we gain a valuable tool which can be used by health service . Therefore, the review presented by the Authors can be a valuable source of information and help to improve the treatments received by patients. In my opinion, the topic was presented in an interesting and accessible way, the manuscript contains all the relevant information. It is written in the correct language typical for this type of scientific work in accordance with the requirements set by the journal. In addition, the tables placed in it significantly enrich its value. An interesting solution is the introduction of the Future Directions paragraph. In my opinion the Authors should only extend the abstract to facilitate the search for the topic by those interested.

Round 2

Reviewer 2 Report

the manuscript in its present form may, in my opinion, be considered for publication